# Alteration of Mitochondrial Integrity as Upstream Event in the Pathophysiology of SOD1-ALS

**DOI:** 10.3390/cells11071246

**Published:** 2022-04-06

**Authors:** René Günther, Arun Pal, Chloe Williams, Vitaly L. Zimyanin, Maria Liehr, Cläre von Neubeck, Mechthild Krause, Mrudula G. Parab, Susanne Petri, Norman Kalmbach, Stefan L. Marklund, Jared Sterneckert, Peter Munch Andersen, Florian Wegner, Jonathan D. Gilthorpe, Andreas Hermann

**Affiliations:** 1Department of Neurology, University Hospital Carl Gustav Carus Dresden, Technische Universität Dresden, 01307 Dresden, Germany; rene.guenther@uniklinikum-dresden.de (R.G.); a.pal@hzdr.de (A.P.); vlz3f@virginia.edu (V.L.Z.); marialiehr788@gmail.com (M.L.); mrudulaparab8@gmail.com (M.G.P.); 2Deutsches Zentrum für Neurodegenerative Erkrankungen (DZNE), 01307 Dresden, Germany; 3Dresden High Magnetic Field Laboratory (HLD), Helmholtz-Zentrum Dresden-Rossendorf (HZDR), 01328 Dresden, Germany; 4Department of Integrative Medical Biology, Umeå University, 90187 Umeå, Sweden; chloe.williams@umu.se (C.W.); jonathan.gilthorpe@umu.se (J.D.G.); 5Department of Molecular Physiology and Biological Physics, University of Virginia, Charlottesville, VA 22903, USA; 6German Cancer Consortium (DKTK), Partner Site Dresden, and German Cancer Research Center (DKFZ), 69192 Heidelberg, Germany; claere.vonneubeck@uk-essen.de (C.v.N.); mechthild.krause@ukdd.de (M.K.); 7OncoRay—National Center for Radiation Research in Oncology, University Hospital Carl Gustav Carus Dresden, Technische Universität Dresden, 01307 Dresden, Germany; 8Clinic for Particle Therapy, West German Proton Therapy Centre Essen (WPE) gGmbH, University Medical Centre of Essen, 45147 Essen, Germany; 9Helmholtz-Zentrum Dresden—Rossendorf, Institute of Radiooncology—OncoRay, 01328 Dresden, Germany; 10Department of Radiotherapy and Radiation Oncology, University Hospital Carl Gustav Carus Dresden, Technische Universität Dresden, 01307 Dresden, Germany; 11National Center for Tumor Diseases (NCT), Partner Site Dresden, University Hospital Carl Gustav Carus Dresden, Technische Universität Dresden, 01307 Dresden, Germany; 12Department of Neurology, Hannover Medical School, 30625 Hannover, Germany; petri.susanne@mh-hannover.de (S.P.); norman.kalmbach@googlemail.com (N.K.); wegner.florian@mh-hannover.de (F.W.); 13Department of Medical Biosciences, Clinical Chemistry, Umeå University, 90187 Umeå, Sweden; stefan.marklund@umu.se; 14Center for Regenerative Therapies Dresden, Technical University Dresden, 01307 Dresden, Germany; jared.sterneckert@tu-dresden.de; 15Department of Clinical Sciences, Umeå University, 90187 Umeå, Sweden; peter.andersen@umu.se; 16Translational Neurodegeneration Section, “Albrecht Kossel”, Department of Neurology, University Medical Center Rostock, University of Rostock, 18147 Rostock, Germany; 17Deutsches Zentrum für Neurodegenerative Erkrankungen (DZNE) Rostock/Greifswald, 18147 Rostock, Germany; 18Center for Transdisciplinary Neurosciences Rostock (CTNR), University Medical Center Rostock, University of Rostock, 18147 Rostock, Germany

**Keywords:** *SOD1*, ALS1, mitochondria, live cell imaging, axonal trafficking

## Abstract

Little is known about the early pathogenic events by which mutant superoxide dismutase 1 (SOD1) causes amyotrophic lateral sclerosis (ALS). This lack of mechanistic understanding is a major barrier to the development and evaluation of efficient therapies. Although protein aggregation is known to be involved, it is not understood how mutant SOD1 causes degeneration of motoneurons (MNs). Previous research has relied heavily on the overexpression of mutant SOD1, but the clinical relevance of SOD1 overexpression models remains questionable. We used a human induced pluripotent stem cell (iPSC) model of spinal MNs and three different endogenous ALS-associated *SOD1* mutations (D90A^hom^, R115G^het^ or A4V^het^) to investigate early cellular disturbances in MNs. Although enhanced misfolding and aggregation of SOD1 was induced by proteasome inhibition, it was not affected by activation of the stress granule pathway. Interestingly, we identified loss of mitochondrial, but not lysosomal, integrity as the earliest common pathological phenotype, which preceded elevated levels of insoluble, aggregated SOD1. A super-elongated mitochondrial morphology with impaired inner mitochondrial membrane potential was a unifying feature in mutant SOD1 iPSC-derived MNs. Impaired mitochondrial integrity was most prominent in mutant D90A^hom^ MNs, whereas both soluble disordered and detergent-resistant misfolded SOD1 was more prominent in R115G^het^ and A4V^het^ mutant lines. Taking advantage of patient-specific models of SOD1-ALS in vitro, our data suggest that mitochondrial dysfunction is one of the first crucial steps in the pathogenic cascade that leads to SOD1-ALS and also highlights the need for individualized medical approaches for SOD1-ALS.

## 1. Introduction

Amyotrophic lateral sclerosis (ALS) is a heterogeneous but fatal progressive neurodegenerative disease, typically with an adult-onset that strikes in the prime of life. The hallmark of ALS is an initial focal onset of the progressive loss of motor neurons (MNs) and their associated tracts, resulting in paresis and muscle atrophy. The patient inevitably expires when respiratory muscles become severely affected and the median survival time is 3–4 years from symptom onset. Only two drugs, Riluzole and Edaravone, have shown limited therapeutic benefits, and the basic mechanisms that underlie disease pathogenesis are unknown, acting as a barrier to the development of effective therapies.

In 1993, mutations in the gene encoding copper zinc superoxide dismutase 1 (SOD1) were the first to be identified as a cause of ALS and are responsible for up to 20% of cases of familial ALS [1]. Approximately 220 different mutations in the *SOD1* gene have been reported in ALS patients [2]. With the exception of a few recessively inherited mutations, such as D90A, the majority of ALS-causing mutations are inherited as a dominant Mendelian trait. SOD1-ALS shows extensive phenotypic heterogeneity, which is unexplained [3]. While the mean age of onset in D90A and A4V cases is similar (47 years), the median survival time from the onset of paresis is 14 years for D90A^hom^ but only 1.1 years for A4V^het^, and these two mutations may be considered as being located at opposing ends of the clinical ALS spectrum [4,5]. The A4V mutation is associated with a variable site of onset (i.e., spinal or bulbar) but always shows a predominant effect on the lower motor neuron system [3,6]. In contrast, the D90A mutation can cause ALS as a homozygous or, rarely, as a heterozygous trait. In the prevalent D90A^hom^ form, ALS is associated with the insidious onset of a slowly progressing phenotype with early clinical and neurophysiological signs of upper motor neuron and long tract involvement [7]. A mutation with a more typical ALS phenotype is R115G, the most prevalent mutation in Central Europe.

SOD1 is a ubiquitously expressed enzyme catalyzing the dismutation of the superoxide radical 2O_2_^−^ + 2H^+^ → H_2_O_2_ + O_2_. Superoxide and H_2_O_2_ are reactive oxygen species (ROS), which are generated as by-products of mitochondrial oxidative phosphorylation [8]. Excessive ROS can result in oxidative stress, and this is one of the commonly proposed mechanisms in the pathogenesis of ALS [9,10]. ROS react indiscriminately with proteins, lipids, polysaccharides and nucleic acids and thereby can interfere with multiple cellular processes, induce inflammatory pathways, excitotoxicity, protein aggregation, organelle stress and eventually cell death [9,11]. Increased ROS levels have been identified in different SOD1 mutant cell lines [12,13]. Because of their high energetic/metabolic demand for oxygen, together with their life-long postmitotic state, neurons are thought to be particularly vulnerable to oxidative stress. Hence, ROS are a therapeutic target for ALS, and Edaravone (an antioxidant) was approved in 2017 by the FDA [14]. However, a reduction in the antioxidant activity of SOD1 is not the only pathogenic pathway involved in ALS, since at least 7 ALS-associated SOD1 mutations have preserved enzymatic activity [5,15]. Interestingly, a complete loss of SOD1 enzymatic activity does not lead to ALS but to a severe, progressive, infantile-onset motor neuron disease in humans [16], presumably due to a perturbation of the signaling function of superoxide/H_2_O_2_ [8,17]. However, it is unknown if dysregulated mitochondrial metabolism acting as a source of perturbed ROS is a key event in the pathogenesis of SOD1-ALS.

Mitochondria are essential for the maintenance of the very long axons of MNs and may be key sites of MN vulnerability in ALS. Apart from maintaining ROS homeostasis, mitochondria are the primary site of ATP production and also important for calcium homeostasis. Mitochondrial transport is perturbed in MNs cultured from SOD1 overexpressing mouse models of ALS and in cortical neurons transfected with mutant SOD1 [18]. In mutant SOD1 mouse models, a noticeable change in mitochondrial morphology has been observed. Depending on the type of mutation, disease duration and cell compartment, highly elongated or fragmented mitochondria have been found [19]. Fusion and fission processes retain the mitochondrial network in a healthy state. Fusion requires the merging of the inner as well as the outer membranes of two mitochondria. This mechanism can compensate for minor mitochondrial defects, or mitigate the effects of environmental damage by sharing mitochondrial components, and can increase oxidative capacity in response to stress [20]. Fission serves as the initial step of mitophagy, an event required to maintain the mitochondrial network by removing crippled mitochondria and to counterbalance excessive mitochondrial elongation [20,21]. Changes in mitochondrial morphology, such as swelling or enlargement, can block their transport in axons [22] and were found in motor nerve terminals in tissue from ALS patients [23,24]. Mitochondrial dysfunction is not limited to SOD1-ALS, but was reported for other ALS causing mutations as well [25]. As in other proteinopathies (Alzheimer’s disease, Parkinson’s disease, Huntington’s disease), a key histological hallmark of ALS is the occurrence of inclusion bodies in affected neurons [26,27,28,29]. Remarkably, inoculation of SOD1 aggregates, prepared from spinal cords of transgenic mice as well as humans carrying mutant SOD1s, was recently shown to transmit both SOD1 aggregation and fatal motor neuron disease to transgenic mice [30,31]. These findings suggest that prion-like effects of mutant SOD1 aggregates are significant contributors to SOD1-ALS, but it remains unknown whether or not prion-like mechanisms are an initiating event in ALS or are perhaps a downstream consequence of pathogenic protein aggregation.

DNA damage is also a pathogenic hallmark of ALS, as well as other neurodegenerative diseases [25,32]. Genomic instability due to loss of function of nuclear SOD1, along with augmented ROS have been proposed as a pathogenic mechanism contributing to ALS [33]. Recent evidence has proposed a nuclear function for SOD1 under oxidative stress by regulating the activation of pathways required for enhanced resilience towards oxidative DNA damage [34]. However, it remains unclear whether DNA damage occurs as an upstream causative event, or as a downstream consequence in the pathophysiology of SOD1-ALS (e.g., due to increased ROS levels) [33].

Many studies of the mechanisms of SOD1-ALS have been based on cell and animal models overexpressing mutant SOD1. More clinically relevant disease models with endogenous levels of SOD1 expression are necessary in order to accurately define the role of SOD1 in the pathogenesis of ALS, especially in the context of protein aggregation. We recently identified hypoexcitability, attenuated sodium channel expression, elevated endoplasmic reticulum stress and increased MN cell death as phenotypes in patient-derived iPSC MNs from SOD1 ALS patients [35]. However, the mechanism(s) underlying these phenotypes remains unclear. To better define the pathophysiological cascade of SOD1-ALS, we have now investigated the degree of SOD1 disorder and aggregation together with mitochondrial integrity, components of the axonal transport machinery and DNA damage in iPSC-derived MNs from patients with three different pathogenic SOD1 mutations. MNs were compartmentalized in microfluidic chambers (MFCs) that enable the study of aligned axons projecting within the microgroove channels with defined retro- vs. anterograde directionality [25,36]. This setup recapitulates the retrograde “dying-back” of spinal ALS MNs in vitro and is a novel method to study early ALS pathogenesis [25].

## 2. Materials and Methods

### 2.1. Patient Characteristics

We studied iPSC-derived spinal MN cultures from ALS patients with the following pathogenic SOD1 mutations: A4V^het^ (p.Ala5Val), D90A^hom^ (p.Asp91Ala) and R115G^het^ (p.Arg116Gly), and compared them to MNs carrying human SOD1 wild type (SOD1 Wt) in four cell lines from healthy volunteers and one gene-corrected isogenic control line of D90A^hom^ (SOD1 D90A igc). All cell lines were obtained from skin biopsies of patients or healthy volunteers and have been described before [25,35,37,38] (Table 1). The performed procedures were in accordance with the Declaration of Helsinki (WMA, 1964) and approved by the Ethical Committee of the Technische Universität Dresden, Germany (EK 393122012 and EK 45022009) and the University of Umeå, Sweden (14-137-31M). Written informed consent was obtained from all participants including for publication of any research results.

### 2.2. Karyotyping

SOD1 iPSC and control cell lines were karyotyped using the HumanCytoSNP-12 v2.1 BeadChip Kit (Illumina Inc., San Diego, CA, USA). All clones showing pathological SNPs were excluded prior to the study.

### 2.3. Genotyping

DNA from the cell lines D90A^hom^ (p.Asp91Ala) and R115G^het^ (p.Arg116Gly) were genotyped by a diagnostic human genetic laboratory (CEGAT, Tübingen, Germany) and the A4V^het^ (p.Ala5Val) cell lines at Umeå University, Sweden. Control lines were also genotyped and verified as SOD1 wild type and also lacking C9ORF72, FUS or TDP43 mutations.

### 2.4. Mycoplasma Testing

All fibroblast lines were checked for *Mycoplasma* sp. prior to and after reprogramming and afterwards, and routine checks for *Mycoplasma* were conducted every three to six months. We used the Venor^®^ GeM Classic mycoplasma detection kit according to manufacturer’s instructions (Cat. No. 11–1025, Minerva Biolabs GmbH, Berlin, Germany).

### 2.5. Generation, Gene Editing and Differentiation of Human iPSC Cell Lines to MNs in Microfluidic Chambers (MFCs)

The generation and expansion of iPSC lines from healthy control and familial ALS patients with defined mutations in the *SOD1* gene (Table 1) were recently described [35,37]. The gene corrected isogenic control to the homozygous mutant SOD1 D90A^hom^ (SOD1 D90A^hom^ igc, Table 1) was generated by CRISPR-Cas9n-mediated gene editing and was fully characterized [37]. The subsequent differentiation to neuronal progenitor cells (NPC) and further maturation to spinal MNs was described previously [25,42]. The coating and assembly of MFCs (Xona Microfluidics RD900) to prepare for the seeding of MNs was performed as described [25,36,38]. MNs were seeded for maturation into one side of an MFC to obtain a fully compartmentalized culture with proximal somas and their dendrites being physically separated from their distal axons, as only the latter type of neurite was capable to grow from the proximal seeding site through a microgroove barrier of 900 µm-long microchannels to the distal site (shown in details in [43]). All subsequent imaging in MFCs was performed at day (D)21 of axonal growth and MN maturation (D0 = day of seeding into MFCs).

### 2.6. Live Imaging in MFCs

Time-lapse movie acquisition was performed as described [25,36]. In brief, to track lysosomes and mitochondria, cells were double-labelled with Lysotracker Red DND-99 (Molecular Probes Cat. No. L-7528) and Mitotracker Deep Red FM (Molecular Probes Cat. No. M22426) at 50 nM each. Trackers were added directly to culture supernatants and incubated for 1 h at 37 °C prior to live imaging. 

### 2.7. Organelle Tracking and Shape Analysis

Recently, we have published a comprehensive description of the automated analytical pipeline starting from object recognition in raw movie data to final parametrization of organelle motility and morphology [36]. In brief, organelle recognition and tracking was performed with the FIJI Track Mate plugin, organelle shape analysis with our custom-tailored FIJI Morphology macro that is based on the FIJI particle analyzer. Both Track Mate and particle analyzer tools returned the mean speed and track displacement for each organelle type (mito- versus lysotracker-labeled) along with the length of mitochondria and area of lysosomes. Subsequent data mining of individual per-movie result files was performed in KNIME to assemble final results files with annotated per-organelle parameters, thereby allowing all data from each experimental condition to be pooled (e.g., all data for mitotracker or lysotracker for a given cell line). In addition, the percentage of moving tracks, defined as the percentage of all tracks with a minimum track displacement of 1.2 µm as an arbitrary threshold for moving as opposed to stationary organelles, was calculated as a post-processing parameter in KNIME. Data were either displayed as bar histogram and scatter dot plot of individual organelles (e.g., mitochondria length) pooled from at least three independent experiments or data were averaged per experiment and displayed as box plot (i.e., each data point presents the mean of one independent experiment).

Analysis of mitochondrial membrane potential with Mitotracker JC-1 (Molecular Probes Cat. No. M34152) was performed as described [25]. In brief, object segmentation was performed with the channel of higher intensity (most often red emission) to generate a selection limited to mitochondria using a custom-tailored FIJI macro. The resulting selection was saved as a region of interest (ROI) and applied to both channels to reveal the total integral intensity and area of mitochondria and background in both channels using the “Measure” command. After area normalization and background subtraction, ratios of integral red/green intensity were taken as mean membrane potential per movie (first frame only) and batch-analyzed in KNIME as for the tracking analysis (see above). The resultant ratios were averaged per experiment and displayed as box plots on a log scale.

### 2.8. Stress Granule Induction and Protein Aggregation

Induction of stress granules and stimulation of protein aggregation was initiated by different treatments: (i) a short term induction of stress granules by incubation with 200 µM sodium arsenite (Cat. No. S7400, Sigma-Aldrich, St. Louis, MO, USA)) for 1 h [44], (ii) a long term treatment (24 h) with a combination of 2.5 mg/mL of the polysome inhibitor puromycin (Cat. No. 58-58-2, InvivoGen Europe, Toulouse, France) and 40 µM of the Hsp70 inhibitor VER155088 (Cat. No. SML0271, Sigma-Aldrich, St. Louis, MO, USA), which blocks stress granule disassembly and chaperone mediated autophagy [44,45], and (iii) heat stress (43 °C for 2 h) in combination with 40 µM VER155088 [46]. For induction and detection of aggresome formation, we used the PROTEOSTAT^®^ Protein aggregation assay kit (Cat. No. ENZ-51035-0025, Enzo Life Sciences, Inc., Farmingdale, NY, USA), which included incubation with 10 µM MG-132 (Cat. No. BML-PI102-0005, Enzo Life Sciences, Inc.), a proteasome inhibitor, for 18 h [45].

### 2.9. DNA Damage

Double strand breaks (DSB) were induced by incubation with 2 µM etoposide (Cat. No. E1383, Sigma-Aldrich, St. Louis, MO, USA) for 1 h. Cells were immediately fixed afterwards and immunocytochemistry was performed. Anti-γH2A.X (1:500, Cat. No. 05-636, Sigma-Aldrich, St. Louis, MO, USA) and anti-53BP1 (1:1000, Cat. No. NB100-304, Novus Biologicals, Centennial, CO, USA) double-positive intranuclear punctae were counted as DSBs. Cells were counted as viable if no signs of nuclear blebbing or fragmentation were detected by Hoechst 33342 (Cat. No. 62249, ThermoFisher Scientific, Waltham, MA, USA) staining as an indication of apoptosis. Ratio of vital cell count before and after 2 µM etoposide treatment was used as a survival readout. For DSB recovery experiments, an induction with 5 µM etoposide for 1 h was performed, followed by a washing step with maturation medium and exchange with fresh maturation medium that did not contain etoposide. After 24 h, cells were fixed and stained for the above mentioned antibodies. Induction of DSBs by X-ray irradiation (Yxlon Maxishot 200 Y.TU/320-D03, Yxlon Int. GmbH, Hamburg, Germany; 200 kV, 20 mA; 0.5 mm Cu filter; dose rate 1 Gy/min) was carried out in a 6-well format. Following irradiation, cells were detached with Accutase (Cat. No. A6964, Sigma-Aldrich, St. Louis, MO, USA) and stained for DNA DSBs with the anti-histone yH2A.X antibody, and yH2A.X-positive cells in irradiated samples relative to untreated control cells were analyzed by flow cytometry (FACSCanto™ II Clinical Flow Cytometry System, BD Bioscience, Franklin Lakes, NJ, USA).

### 2.10. Immunofluorescence Stainings

For immunofluorescence staining, classical protocols were used. Primary antibodies were incubated overnight at 4 °C, except for the γH2A.X antibody (2 h at room temperature). The following primary antibodies were used: to detect stress granules we used the antibody anti-EIF3ƞ (1:1000, Cat. No. sc-16377, Santa Cruz Biotechnology Inc., Dallas, TX, USA); natively folded SOD1 was detected with the anti-SOD1 (1:300, Cat. No. CB14379, Cell Applications Inc, San Diego, CA, USA) antibody; to detect aggregates of misfolded SOD1 we used an anti-SOD1 peptide sequence antibody raised against amino acid 58 to 72 (SOD1 aa 58–72, 1:500, Cat. No. AS13 2644, Agrisera, Vännäs, Sweden). To test the specificity of the peptide sequence antibody, we used a previously established HeLa cell model overexpressing a green fluorescence protein-tagged SOD1 mutant [46]. To detect DSBs we used anti-histone γH2A.X (1:500, Cat. No. 05-636, Sigma-Aldrich, St. Louis, MO, USA) and anti-53BP1 (1:1000, Cat. No. NB100-304, Novus Biologicals, Centennial, CO, USA). Nuclei were counter stained using Hoechst 33342 (Cat. No. 62249, ThermoFisher Scientific, Waltham, MA, USA).

### 2.11. Image Quantification

For immunofluorescence, a minimum of 3 independent experiments based on 3 separate differentiation pipelines were performed. Twenty images per experiment of mature MNs at D21 after seeding into MFCs, or of uncompartmentalized NPCs, were analyzed as described above. For movie analysis of MFCs (organelle tracking and shape, mitochondrial membrane potential), at least ten movies were acquired of each MFC (=one technical replicate) with three MFCs per experiment and at least 4 independent experiments (=MN differentiation pipeline) per cell line.

### 2.12. Measurements of ATP in MNs

Measurements of relative ATP concentration using an A-team sensor and FLIM imaging are described in detail in Zimyanin et al. (submitted). Briefly, MNs grown in microfluidic chambers were infected with lentivirus that carry either WT or mutant (R122K, R126K) non-sensing A-team ATP-FRET sensors (Imamura et al., 2009). MNs were infected with carrier lentivirus during maturation stage at the last re-plating or 1 day after re-plating. Cells expressing full constructs were detected by sensor fluorescence in both CFP and YFP channels. For FLIM imaging we used an inverted laser scanning confocal microscope (Zeiss LSM 780/FLIM microscope) in the imaging facility of BIOTEC in Dresden with a temperature and CO_2_ controlled chamber.

The microscope was equipped with a Becker & Hickl dual channel FLIM unit and used a 440 nm pulsed diode laser. For A-team imaging, a CFP/YFP double filter cube set F46-001 and 40×/1.2 LD LCI Plan-Apochromat lens was used. FLIM data fitting is based on the B&H handbook and software. The offset and scattering were set to 0. Shift was optimized to make sure that Chi2 was close to 1 (between 0.7 and 2).

FLIM processing and analysis was performed following [47]. ImageJ/FIJI custom plugin described in [47] was used for cell segmentation to create single pixel locations by X-Y coordinates, specific for cytosolic or axonal regions in MNs, excluding the nucleus, background and noise measurements. Those locations were then applied to the FLIM data to extract FLIM parameters for each pixel. A custom Python code was used to analyze different data combinations to produce ratios, means, medians, and histograms, further charted and statistically analyzed in GraphPad Prism.

For mitochondrial inhibitor treatments, oligomycin A 10 µM (10 mM stock) and CCCP 10 µM (10 mM stock) were added into the cell culture medium 1 h prior to imaging.

### 2.13. Cell Extracts SOD1 Analysis

Cell pellets were thawed rapidly in a water bath at 25 °C for 1 min and then lysed in ice-cold PBS containing Complete EDTA-free protease inhibitor cocktail (Cat. No. 11 873 580 001, Sigma-Aldrich, St. Louis, MO, USA), 40 mM iodoacetamide (IAM, Cat. No. I1149, Sigma-Aldrich, St. Louis, MO, USA) and 0.5% (*v*/*v*) Nonidet P-40 (NP-40, Cat. No. 11754599001, Sigma-Aldrich, St. Louis, MO, USA) using a Sonifier Cell Disrupter (Branson, Danbury, CT, USA). Lysates were centrifuged at 20,000× *g* for 30 min at 4 °C and the protein content of the cell lysate supernatant was determined using the Pierce™ BCA Protein Assay Kit (Cat. No. 23225,ThermoFisher Scientific, Waltham, MA, USA). Disordered SOD1 was quantified in freshly prepared extracts by misELISA (see below). Pellets were snap frozen on dry ice and stored at −80 °C for analysis of SOD1 in the detergent-insoluble fraction (see below).

### 2.14. Quantification of Disordered and Total SOD1 by ELISA

Disordered and total SOD1 were quantified in cell extracts using sandwich ELISAs described previously. This method has been extensively validated in both patient-derived fibroblasts and iPSC-derived MNs [48,49,50].

### 2.15. Quantification of SOD1 in Detergent-Insoluble Aggregates

The amount of full length SOD1 in the detergent-insoluble fraction was quantified by western blotting as described previously [48]. The amount of aggregated SOD1 in the detergent insoluble fraction was expressed as a percentage of soluble SOD1 present in the corresponding cell extract, determined by total SOD1 ELISA [48].

### 2.16. Statistical Analyses of ELISA and Western Blotting

Statistical analyses were performed using GraphPad Prism version 5.01 (Graphpad Software Inc, San Diego, CA, USA) and Statistica 13.2 (StatSoft Europe GmbH, Hamburg, Germany). To test for significance between multiple groups, one-way ANOVA followed by Bonferroni’s multiple comparisons test was used. Alpha < 0.05 was used as the cut off for significance (* *p* < 0.05, ** *p* < 0.005, *** *p* < 0.001, **** *p* < 0.0001). All values are given as mean ± SD.

## 3. Results

### 3.1. Aggregates of SOD1 Are Not Detected in iPSC-Derived MNs from SOD1-ALS Patients

A pathological hallmark of ALS is the presence of intracellular inclusion bodies containing insoluble protein aggregates. We first investigated whether or not insoluble aggregates of SOD1 are present in ALS patient-derived MN models that do not overexpress mutant SOD1. Wild-type SOD1 was ubiquitously distributed and could be detected in the cytosol, dendrites and axons and to a lesser extent in the nucleus (Figure 1A). We did not find any inclusions that immunostained positively for disordered SOD1 in unstressed MN cell cultures. Stress granules are recognized precursors of aggregate formation, especially for TDP-43 and FUS mutant ALS [51]. Although SOD1 has been reported to interact with stress granule components in SOD1 overexpressing cell cultures and to colocalize with SOD1-positive inclusions in the spinal cord of SOD1 mouse models, SOD1 does not appear to be an RNA-binding protein [52,53]. To investigate a possible link between stress granules and SOD1 in patient-derived MNs, we induced stress granule formation with arsenic acid, heat stress or puromycin (Puro) in combination with heat shock protein 70 (Hsp70) inhibition (VER155088, Ver) (Figure 1A). MG132, a proteasome inhibitor, was used to induce aggresome formation (Figure 1B). We investigated inclusion formation using an antibody raised against the folded SOD1 protein (totalSOD1) as well as with a SOD1 peptide sequence antibody (SOD1, raised against amino acids 58–72), which exclusively binds to the disordered apo SOD1 protein, but not to holo SOD1 [54]. All tested conditions showed either stress granules (by ElF3n marker) or aggresome formation, but both formations were negative for total or disordered/misfolded SOD1 (Figure 1A,B), thereby indicating that our immunostaining did not detect SOD1-positive inclusions upon stress. As a positive control for SOD1 aggregation, we transfected HeLa cells with GFP-tagged A4V mutant SOD1 [46] and induced SOD1 aggregation with heat stress in combination with Hsp70 inhibition (Figure 1C), or with MG132 (Figure 1D). We found strong colocalization of cytoplasmic SOD1-GFP inclusions and disordered SOD1 antibody staining. In contrast, the totalSOD1 antibody showed no specific staining of cytoplasmic SOD1-GFP inclusions. Taken together, our results show that patient-derived spinal MNs with endogenous levels of mutant SOD1 expression did not exhibit visible SOD1 aggregation.

### 3.2. Augmented Levels of Disordered SOD1 upon Stress in SOD1 Mutant MNs

Native SOD1 is a rigidly folded homodimer and this conformation is essential for enzymatic function. Polyclonal anti-peptide antibodies that specifically recognize epitopes that are exposed in human SOD1 when the native conformation is lost have been developed and used in an ELISA format for quantification of even minute amounts of disordered SOD1 [49,55,56], as well as in extracts from patient-derived fibroblasts [57] and iPSC-derived MNs and astrocytes [48]. Disordered SOD1 occurs under physiological conditions but the amount is marginal compared to the native counterpart due to refolding, or to rapid degradation by the proteasome [48]. However, mutations in SOD1 and stress responses, i.e., the occurrence of aggresomes and stress granules, can lead to increased accumulation of disordered SOD1 as a pathological hallmark [48,49]. Thus, we wished to investigate whether increased occurrence of disordered SOD1 could be detected as an early event in SOD1-ALS by the more sensitive misELISA method. In order to distinguish between soluble and aggregated SOD1, we prepared soluble and detergent-insoluble fractions from cell lysates. First, we determined the total amounts of soluble SOD1 in all lines using antibodies that recognizes all SOD1 species, irrespective of conformational state. We found reduced levels of soluble total SOD1 in all untreated mutant lines (D90A, A4V, R115G) compared to isogenic and healthy controls (D90A igc, Ctrl, Figure 2A, blue bars). Neither induction of stress granule formation with Ver and Puro (Figure 2A, compare blue vs. green bars) nor inhibition of proteasome function using MG132 (Figure 2A, compare blue vs. red bars) led to alterations in soluble total SOD1 in any line. In contrast, we found elevated levels of disordered SOD1 in the soluble fraction in untreated samples carrying unstable SOD1 A4V and R115G mutations as compared to control MNs, but not in the line carrying the stable SOD1 D90A^hom^ variant (Figure 2B, blue bars). Combined treatment using Ver + Puro did not impact on disordered SOD1 in any line except Ctrl1 (Figure 2B, compare blue vs. green bars), but MG132 led to a further increase in A4V and R115G (Figure 2B, compare blue vs. red bars). When analyzing the detergent-insoluble fraction by western blotting, we detected low but very similar amounts of aggregated SOD1 in all untreated lines (Figure 2C). Upon treatment with MG132, aggregated disordered (misfolded) SOD1 exhibited augmented levels in A4V and R115G (Figure 2C,D) but not in D90A. Consistent with observations in the soluble fraction, stress granule induction (Ver + Puro) did not impact on SOD1 aggregation in the detergent-insoluble fraction (Figure 2C,D). In summary, aggregated SOD1 was undetectable in unstressed mutant SOD1 MNs, while soluble disordered SOD1 was increased in MNs carrying the unstable A4V and R115G mutants (Figure 2B). A4V and R115G SOD1 mutants showed increased aggregation upon proteasome inhibition (Figure 2C,D). Importantly, we found no evidence for SOD1 being recruited to SG (Figure 1A,B) or that stress granule induction (VER + Puro) led to increased soluble disordered (Figure 2B) or detergent-resistant SOD1 (Figure 2D).

### 3.3. No Signs of Elevated DNA Damage or Impaired DNA Repair in SOD1 Mutant MNs

DNA stability is an important aspect of the pathophysiology of ALS [25]. Genomic instability, due to loss of functional SOD1 in the nucleus along with increased levels of ROS, was recently described [33], which led to the activation of genes crucial for enhanced resilience towards oxidative DNA damage [34]. Thus, we studied the appearance of spontaneous DNA-damage in SOD1 mutant MNs as well as their ability to deal with DNA damaging events. Neither basal levels of DNA double strand breaks (DSBs) (Figure 3A,B), nor induction and recovery after induction of DSBs through etoposide treatment (Figure 3A,C), nor induction of DNA damage after X-ray irradiation (Figure 3D,E) significantly differed between SOD1 mutants and SOD1 wildtypes. Furthermore, survival of MN after DNA damage in SOD1 mutants was indistinguishable from wild type controls (Figure 3F). These data argue against DNA damage as a principal mechanism for SOD1 mediated-ALS, but this does not exclude them as later secondary events downstream in the pathophysiological cascade.

### 3.4. Elongated Mitochondrial Morphology in SOD1 Mutant MNs

We recently demonstrated increased cell death in iPSC-derived MNs carrying mutant SOD1 [35]. Mitochondrial homeostasis is necessary for the vitality and viability of neuronal cells. Furthermore, genome wide analysis of SOD1 MNs mainly revealed alterations in the metabolic pathways and neuroactive-ligand–receptor interactions [58]. Thus, we hypothesized that an impairment of mitochondrial homeostasis plays a major role in the pathogenesis of SOD1-ALS. We first analyzed the morphology of mitochondria in the different SOD1 Mt lines in comparison to SOD1 Wt including an isogenic control (D90A igc) of the homozygous SOD1 D90A mutant (D90A). In total, over 50,000 mitochondria were analyzed semi automatically for their morphology in images acquired from axons of live MNs cultured in compartmentalized MFCs (Figure 4A,B). We verified that the merging of data from the distal and proximal readout positions in MFCs on the one hand and the pooling of all SOD Wt control lines (pooled Ctrl: Ctrl1, Ctrl2, D90A igc, Table 1) on the other hand was a valid approach (Appendix A), as data obtained under these conditions were statistically indistinguishable from each other. Mean mitochondrial length was significantly increased in SOD1 Mt (pooled SOD1 Mt, 2.07 ± 0.44 µm) as compared to SOD1 Wt (pooled Ctrl, 1.55 ± 0.13 µm) (Figure 4C). SOD1 R115G (2.10 ± 0.41 µm, *p* < 0.01) and SOD1 D90A (2.15 ± 0.48 µm, *p* < 0.0001) had significantly longer mitochondria in comparison to SOD1 Wt (pooled Ctrl), whereas SOD1 A4V (1.75 ± 0.11, *p* = 0.33) showed no significant difference in mean mitochondrial length in comparison to SOD1 Wt (pooled Ctrl, Figure 4D). The D90A^hom^ line was significantly different to its D90Aigc counterpart (*p* < 0.01), with increased mean length of mitochondria (Figure 4D,G), thereby confirming that the observed mitochondrial phenotypes were induced by the underlying SOD1 mutation and were not due to genetic background variation. Analyzing the distribution of single mitochondrion length revealed, in addition to the increased mean length, a fraction of super elongated mitochondria in SOD1 D90A and R115G (Figure 4E–G, boxed in red). Importantly, this mitochondrial elongation was not observed in neural precursor cells from the same lines (NPCs) (Figure 4A and Appendix A).

### 3.5. Mitochondrial but Not Lysosomal Trafficking Phenotypes in SOD1 MNs

Mitochondrial integrity depends on shuttling within the axon, including proper retrograde transport to perinuclear spaces to fuse for rejuvenation, or autophagic clearance [59]. To test whether the super-elongated mitochondria in mutant SOD1 MNs (Figure 4) occurred concomitantly with perturbed trafficking we analyzed the motility of axonal mitochondria in MFCs. Moreover, to address whether such perturbed trafficking events were specific for mitochondria, or also affected other organelle types, we analyzed lysosomes as well. Of note, there was no difference between the distal and proximal axonal readout positions in MFCs and no significant clone to clone differences were observed; thus, we pooled data from proximal and distal positions as well as from all SOD Wt lines for further analysis (pooled Ctrl: Ctrl1, Ctrl2, D90A igc, Table 1) (Appendix A). Automated tracking analysis in MFCs revealed a significantly increased mean proportion of moving mitochondria in the SOD1 Mt (pooled SOD1 Mt, 17.4 ± 6.5%, *p* < 0.05) as compared to SOD1 Wt (pooled Ctrl, Figure 5A). In detail, SOD1 D90A (17.8 ± 6.4%, *p* < 0.05) and SOD1 R115G (21.3 ± 4.9%, *p* < 0.01) had a significantly higher mean proportion of moving mitochondria as compared to SOD1 Wt (Figure 5B) (pooled Ctrl, 12.9 ± 3.4%), whereas SOD1 A4V (10.96 ± 4.4, *p* = 0.5) was not different (Figure 5B). Mitochondrial mean speed (pooled Ctrl: 0.54 ± 0.05 µm/s; pooled SOD1 Mt: 0.58 ± 0.05 µm/s; *p* < 0.05) was moderately but significantly increased in a group comparison (Figure 5C) but not in the individual SOD1 Mt line comparison (Figure 5D). Mean mitochondria track displacement (pooled Ctrl: 2.60 ± 0.55 µm; pooled SOD1 Mt: 2.56 ± 0.34 µm; *p* = 0.7) and the mean fraction of mitochondria moving anterogradly (pooled Ctrl: 50.37 ± 6.91%; pooled SOD1 Mt: 51.41 ± 4.57%; *p* = 0.6) was not different between groups (Figure 5E,F and Appendix A), thereby excluding a directional movement bias due to the underlying mutation.

Interestingly, neither lysosomal motility (Figure 6A–C,E) nor lysosomal size (Figure 6D) was affected, thereby suggesting a specific impact on mitochondria, arguing against a systemic alteration of cell organelle motility behavior including axon skeleton and/or molecular motors.

### 3.6. Hypopolarization of Mitochondria in SOD1 Mt MNs

Mitochondrial fusion can compensate for minor mitochondrial defects, mitigate the effects of environmental damage by sharing mitochondrial components and increase oxidative capacity in response to toxic stress [20]. Thus, we investigated whether or not the inner membrane potential—known to be an important indicator of proper mitochondrial function—is disturbed in lines carrying mutant SOD1s. To this end, we used JC-1 as a marker for mitochondrial inner membrane potential relevant for mitochondrial integrity. Both SOD1 Mt cell lines tested (D90A, A4V) (pooled ratio 2.02 ± 1.13) had a dramatically reduced membrane potential in comparison to SOD1 Wt (pooled ratio 10.2 ± 0.81), including a significant difference between the isogenic control and its respective D90A mutant line (Figure 7 and Appendix A). Once more, no difference between proximal and distal axons was observed.

### 3.7. SOD1 Mt MN Have Reduced Levels of ATP

Changes in shape, motility and hypopolarization of mitochondria in SOD1 mutant MNs should disrupt their metabolic function significantly and lead to a reduction in cellular ATP. We have used the Förster resonance energy transfer (FRET)-based ATP biosensor A-team as recently described by [47,60] to establish a robust and efficient way to measure relative changes in ATP concentration in individual neurons and their subcompartments. In this approach, binding of ATP by the sensor leads to its conformational change and an increase in FRET between donor (CFP) and acceptor (YFP) subunits. To avoid artifacts associated with measurements of fluorescent intensity-based FRET, we utilized Fluorescence Lifetime Imaging (FLIM) approaches [47,60]. FLIM microscopy measures multiple single-photon fluorescence events and allows the building of a histogram, and therefore an estimation of fluorescence lifetime and amplitude (i.e., number of detected photons) for each fluorophore. Since FRET decreases mean lifetime of the donor (T_m_), one can quantify the extent to which FRET occurs, provided the donor lifetime without FRET is known, and therefore estimate relative changes in ATP concentrations. Using this approach, we tested whether SOD1 D90A^hom^ mutant motoneurons with the broadest mitochondrial phenotypes also have a detectable reduction in ATP levels in their somata and axons (Figure 8).

As our baseline, we used lifetime of the CFP donor in a R122K/R126K mutant isoform of ATeam1.03 that has no detectable ATP binding and therefore no ATP-dependent FRET events. Mutant A-team isoform has much higher lifetime values in comparison to fully functional A-team sensor in wildtype motoneurons (T_m_
^WT soma^ = 1379 ± 30.18 ps (SEM), *n* = 35 versus T_m_
^WT ATeam1.03 R122K/R126K^ = 1732 ± 23.24 ps, *n* = 26, *p* < 0.0001) and thus provides a relative range of ATP concentration measurements in our cells. Inhibition of oxidative phosphorylation with two commonly used and well characterized mitochondrial inhibitors CCCP and Oligomycin A also significantly increased the mean lifetime of the CFP donor of the functional sensor (T_m CCCP_ = 1578 ± 28.79 ps, *n* = 24, *p* < 0.0001 and T_m Oligom_ = 1575 ± 36.48 ps, *n* = 11, *p* = 0.0014). These changes indicate that the increase in mean lifetime (T_m_) of the CFP donor in our measurements is due to the reduction of ATP levels in the cytoplasm, suggesting a validity of this approach to estimate relative ATP concentrations in living cells and their subcompartments (Figure 8B,D).

We then compared mean lifetime of CFP-donor in wt and SOD1 D90A mutant MNs and detected a significant increase in both somata (Figure 8B,E) (T_m_
^WT soma^ = 1379 ± 30.18 ps (SEM), *n* = 35 versus T_m_
^SOD1 soma^ = 1545 ± 49.21 ps, *n* = 34, *p* = 0.004) and axons (Figure 8C,F) (T_m_
^WT axons^ = 1522 ± 35.3 ps, *n* = 16 versus T_m_
^SOD1 axons^ = 1693 ± 40.68 ps, *n* = 21, *p* = 0.0044). These measurements would strongly support our hypothesis that changes in mitochondrial dynamics, morphology and polarization in SOD1 mutant MNs lead to a significant reduction of ATP levels, thereby imposing detrimental metabolic consequences for the affected MNs.

## 4. Discussion

After almost three decades following the discovery of the contribution of SOD1 mutation to ALS, the pathogenic mechanisms of mutant SOD1 toxicity are not clearly resolved and identification of the crucial initiating pathways remains elusive. Although gene therapies to reduce mutant SOD1 expression using anti-sense approaches are under development and show promising beneficial effects, a long term therapeutic effect without mitochondrial damage needs to be demonstrated in patients with different *SOD1* mutations, each impacting distinctly on the biophysical properties of the SOD1 protein [61]. The data presented here suggest that there are widely different effects on mitochondrial homeostasis in neurons derived from patients with *SOD1* mutations with distinct properties.

To shed further light on the early pathological aspects of SOD1-ALS disease, we performed a systematic study analyzing commonly proposed pathomechanistic aspects of SOD1 ALS, namely SOD1 misfolding and aggregation, DNA damage, mitochondrial morphological and functional integrity and axonal transport. We utilized human iPSC-derived spinal MNs expressing endogenous levels of three different paradigmatic SOD1 mutations. We identified altered mitochondrial integrity in SOD1 mutants as the first and most prominent pathophysiological change, while systemic axon trafficking seemed not to be perturbed, and augmented DNA damage, as well as large alterations in protein aggregation, were absent. MNs expressing SOD1 R115G and most prominently SOD1 D90A had elongated mitochondria (Figure 4), a higher fraction of moving mitochondria (Figure 5A), a strongly reduced membrane potential compared to SOD1 Wt (Figure 7) and reduced intracellular ATP levels (Figure 8). SOD1 A4V did not show these morphological and motility alterations of mitochondria, although its membrane potential was also strongly reduced (summary Table 2). Thus, the overarching common phenotype in SOD1 mutant MNs is mitochondrial damage.

DNA damage was recently reported to be a key player in neurodegeneration of ALS, and some of the genes (FUS, TARDBP, ANG, TAF15) causing familial forms of ALS are involved in DSB DNA repair [62]. Neurons are susceptible to ROS-induced DNA damage, and SOD1 is the major scavenger for superoxide radicals and, therefore, might prevent cells from ROS-induced DNA damage. Moreover, a non-canonical nuclear function of SOD1 in regulating the expression of oxidative resistance and DNA damage repair genes under oxidative stress conditions was recently described [34]. In contrast to this, despite mitochondrial shape and motility alterations (Figure 4 and Figure 5), nuclear DNA damage was not increased and DNA repair capacity in SOD1 Mt was not compromised (Figure 3), despite a reduction of SOD1 protein levels in all mutant lines (Figure 2A). We recently reported DNA damage as an upstream event in the pathophysiology of FUS-ALS iPSC-derived MNs, in which DNA damage induction phenocopies mutant FUS mitochondrial phenotypes (depolarization, severe hypomotility mainly in distal axons, lysosomal motility mirrored mitochondrial phenotypes). However, these phenotypes were significantly different from the mitochondrial/axonal phenotypes identified here in SOD1 mutant MNs (elongated mitochondria, hypermotility, lysosomes not affected, no region-specific alterations). This suggests that mitochondrial alterations in mutant SOD1 MNs are not downstream of DNA damage or impaired DNA repair. Consistently, DNA damage was not identified in a recent study in a SOD1 mouse model of ALS [63].

Protein instability and increased aggregation rate correlated with decreased survival time of ALS patients [64,65]. Protein aggregation and its clearance is one proposed upstream mechanism in SOD1-ALS [66,67]. However, the majority of previously reported studies addressing protein misfolding and aggregation were performed with cell lines or animal models overexpressing human mutant SOD1 [68,69]. Furthermore, a recent study showed that large SOD1 aggregates do not impact cell viability, suggesting aggregation may be a protective mechanism [70]. In our human iPSC-derived spinal MNs with a patient-specific genetic background that presumably better reflects the background of human disease, we could not detect large, insoluble aggregates of SOD1 in the human spinal MNs even under stress conditions (Figure 1A,B and Figure 2C,D). However, soluble disordered (‘misfolded’) SOD1 could be detected in the A4V and R115G mutant even without stress, and levels further increased after proteasome inhibition with MG132 using sensitive ELISA-based quantification (Figure 2B). Notably, the D90A mutant, which showed the most prominent mitochondrial morphology changes, did not show significantly increased levels of insoluble or soluble disordered SOD1 protein even after MG132 treatment (Figure 2B–D). Additionally, in untreated conditions, we did not find significantly increased levels of disordered SOD1, which is consistent with the near Wt stability of SOD1 D90A protein.

Of note is the fact that different groups have found that inoculation of SOD1 aggregates prepared from spinal cord not only from transgenic mice overexpressing mutant SOD1s but also from SOD-ALS patients carrying mutant SOD1s expressed at endogenous levels are able to transmit both SOD1 aggregation and fatal motor neuron disease to transgenic mice [30,31]. It should also be mentioned that iPSC generation leads to rejuvenation of reprogrammed cells [71], which could explain different results as compared to postmortem tissue. Nevertheless, iPSC model systems are particularly suited to detect early pathophysiological events, arguing once more for mitochondrial damage as an early event in SOD1-ALS, which might even precede SOD1 misfolding and aggregation. This does not exclude a significant role of SOD1 misfolding and aggregation in disease spread and progression.

Even though mitochondrial elongation was not present in A4V lines, A4V also showed significantly decreased membrane potential in SOD1 mutant MN mitochondria, which was as pronounced as in the D90A line. Mutation-dependent differences of disease severity and disease onset are well known among different SOD1 mutants in ALS [65]. Differences in mutation position have a strong effect on the level and properties of the mutant SOD1 protein and cellular compensatory homeostatic mechanisms. Mitofusion and mitofission are in a dynamic balance that maintains mitochondrial function and enables fast responses to environmental cues. Mitofusion or hyperfusion can protect dysfunctional mitochondria from mitofission and macroautophagy by diffusion and sharing of components [20], which could explain the mitochondrial elongation in SOD1 D90A and R115G. Interestingly, treatment with a mitochondrial fission inhibitor and a fusion promoter led to elongated mitochondria and also increased mitochondrial motility, however this was without signs of hypopolarization [36]. Further studies are needed to unravel the cause of mitochondrial elongation in our patient-derived iPSC lines as well as the dispersion of SOD1 misfolding and mitochondrial damage phenotypes.

## 5. Conclusions

Our data suggest mitochondria to be among the first crucial organelles that suffer from and have to compensate for mutant SOD1 in MNs. These phenotypes were abundant before aggregate formation or detergent-resistant SOD1 could be found. SOD1 was neither found in SG nor further disordered after SG induction, whereas the latter was clearly the case when proteasome function was inhibited (Figure 2B–D). Therefore, further investigations are necessary to clarify how mitochondrial integrity and dynamics (i.e., mitofusion and mitofission) can serve as therapeutic intervention targets in mutant SOD1 ALS and how these are connected to SOD1 protein misfolding and aggregate spreading. Finally, our data clearly show distinct mutation-dependent biophysical properties of the SOD1 protein, indicating a potential need for individualized therapeutic strategies along with sustained long term reductions of mutant SOD1 levels, whereas any impairment of mitochondrial function must be avoided.

## Figures and Tables

**Figure 1 cells-11-01246-f001:**
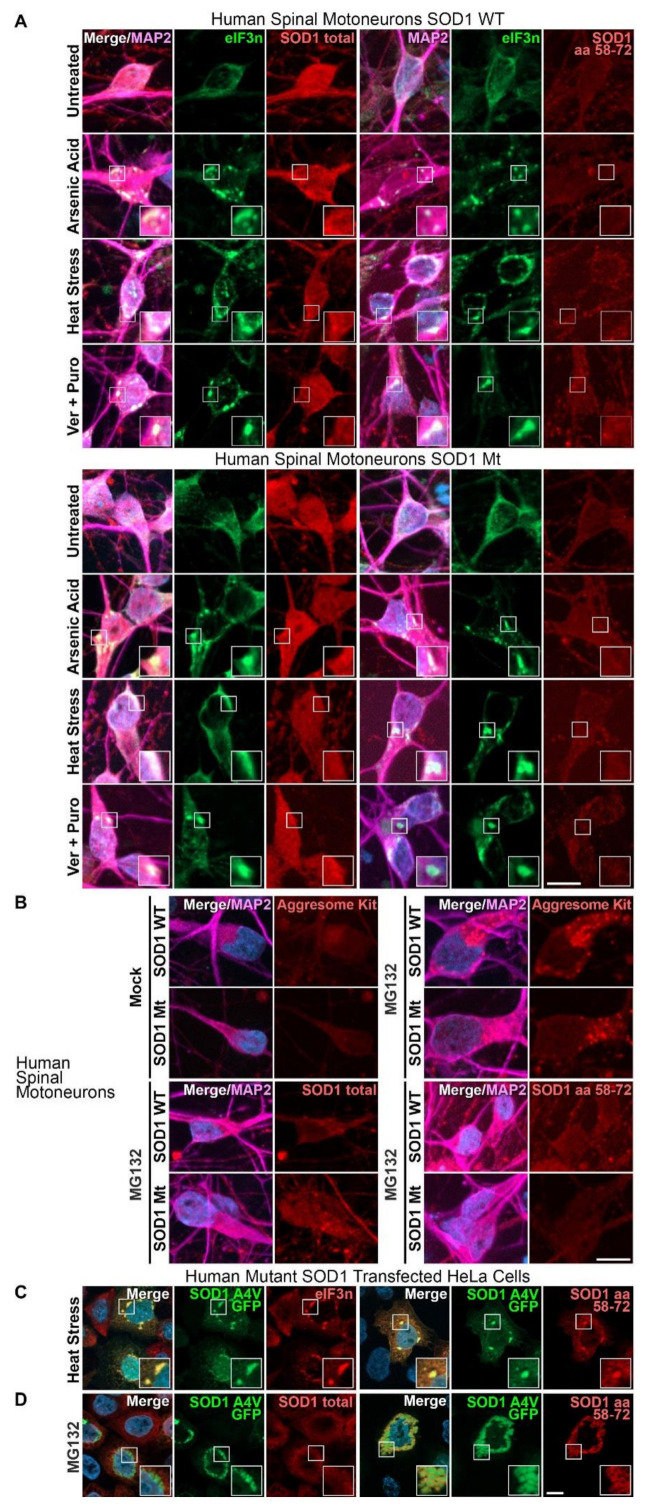
Distinct stress response of endogenous versus overexpressed mutant SOD1. Immunostaining of patient-derived MNs (**A**,**B**) and HeLa cells (**C**), SOD1 WT: MNs from healthy control probands, representative examples from Ctrl1 (Table 1), SOD1 Mt: MNs from mutant SOD1 ALS patients, representative examples from R115G (Table 1). (**A**) Stress granules (eIF3ƞ) could be induced in human MNs with arsenic acid, heat stress and by Ver/Puromycin treatment (Hsp70 and polysome inhibition), but accumulation of natively folded (SOD1 total) and disordered SOD1 (SOD1 aa58–72) was not detected in these stress granules. (**B**) Proteasome inhibition (MG132) led to aggresome formation (Aggresome kit) but not to accumulation of natively folded or misfolded SOD1. (**C**) In contrast, stress granules in a HeLa cell model with human SOD1 Mt overexpression (CMV-promoter-SOD1 4AV-GFP-tagged transfected HeLa cells) were both positive for either the GFP-tag and the peptide antibody reacting with disordered/misfolded SOD1 (SOD1 amino acids 58–72). (**D**) Likewise, proteasome inhibition led to formation of large aggregates of mutant SOD1 in the HeLa overexpression model, which were negative for natively folded SOD1 but strongly reacted with the SOD1 amino acid 58–72 peptide antibody. Scale bars = 10 µm. This experiment served as a positive control in the HeLa cell model, validating the specificity of the SOD1 amino acid 58–72 peptide antibody and verifying the absence of significant amounts of disordered SOD1 in human MNs.

**Figure 2 cells-11-01246-f002:**
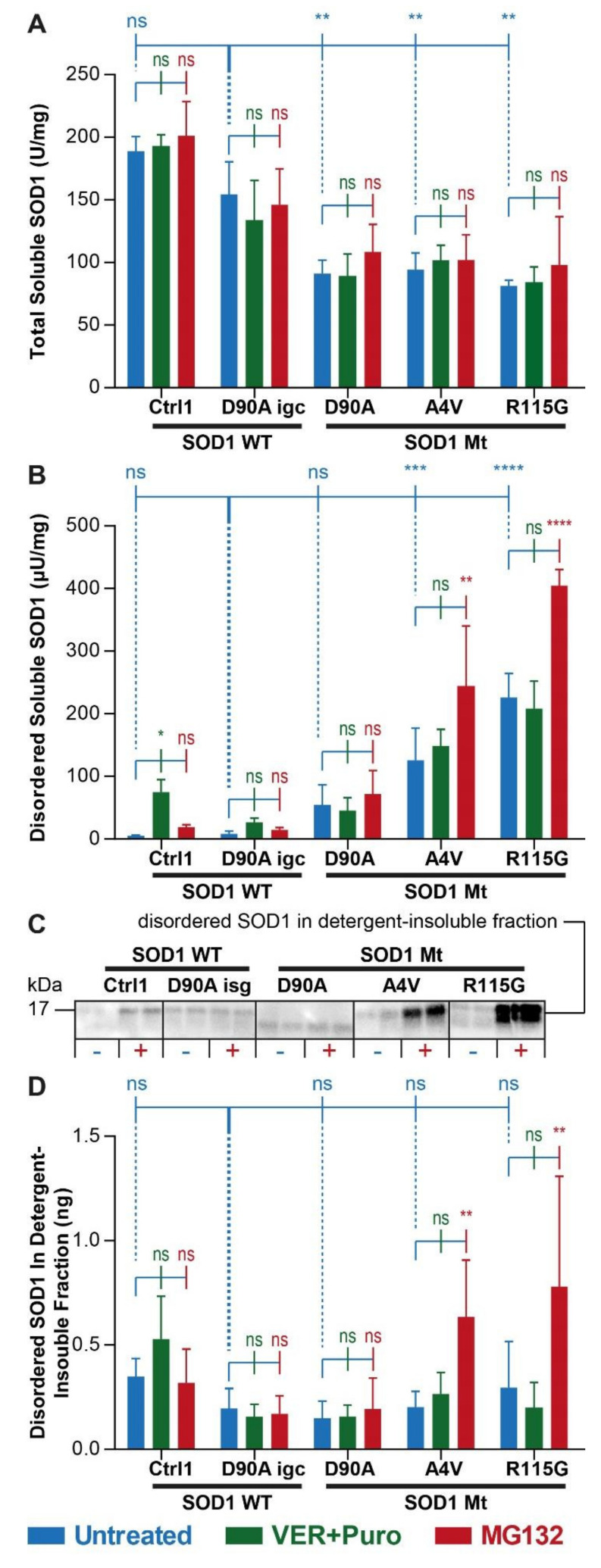
SOD1 misfolding is induced by proteome inhibition but not stress granule assembly in SOD1 mutant MNs. (**A**) Levels of total soluble SOD1 in supernatants of spinal MN extracts by ELISA performed with an antibody recognizing all natively folded and misfolded SOD1 species. Neither VER + Puro (stress granule induction, in green) nor MG132 (proteasome inhibition, in red) treatment altered levels of total SOD1 in any line (significances “ns” in green and red with respect to the untreated line in blue). Levels of total SOD1 were significantly reduced in untreated (in blue) SOD1 Mt D90A, A4V and R115G cells compared to untreated (in blue) SOD1 WT including isogenic gene-corrected control D90A igc cells (significances “**” in blue on top with respect to D90A igc). (**B**) Levels of disordered soluble SOD1 in supernatants of spinal MN extracts by ELISA performed with the peptide antibody against amino acids 24–39 recognizing this epitope only in disordered SOD1. Disordered SOD1 was significantly increased in SOD1 Mt A4V and R115G, but not D90A, after MG132 treatment (in red) but not induced with VER + Puro (in green) as compared to untreated cells (in blue). Both the A4V and R115G, but not D90A, exhibited increased levels of disordered SOD1 already in the untreated state (in blue; significances “**”, “****” in blue on top with respect to untreated D90A igc). (**C**) Western blot analysis of detergent-insoluble pellet fraction of spinal MN extracts performed with antibody against amino acids 24–39 for specific detection of disordered SOD1 in untreated (−) versus MG132-treated (+) cells. All untreated lines exhibited similar minute levels of disordered SOD1. Both the A4V and R115G, but not D90A, exhibited increased levels of disordered SOD1 upon MG132 treatment. (**D**) Quantification of (**C**). MG132 treatment (in red) of A4V and R115G, but not D90, increased the amount of disordered SOD1 in the pellets whereas VER + Puro (in green) did not. Five independent biological replicates for each cell line and each condition. Asterisks: significant alteration with respect to untreated reference in blue by two-way ANOVA with Bonferroni post test. * *p* < 0.05, ** *p* < 0.01, *** *p* < 0.001, **** *p* < 0.0001, ns not significant.

**Figure 3 cells-11-01246-f003:**
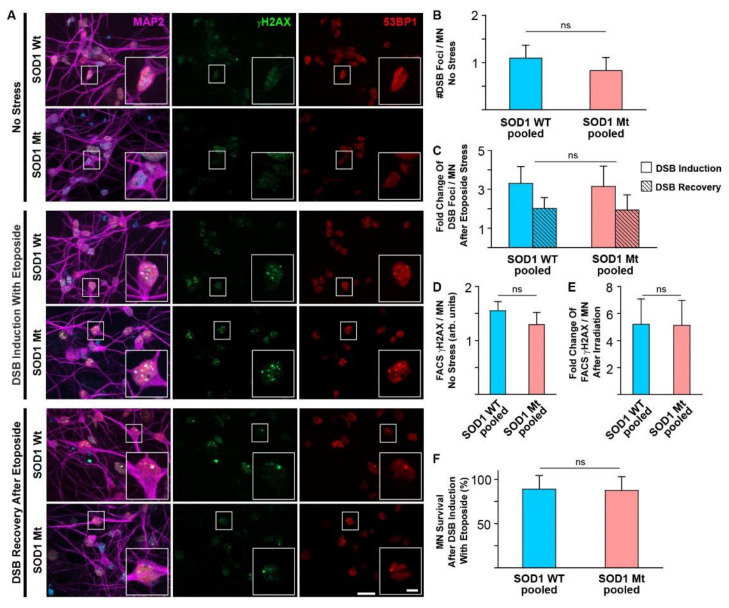
DNA damage is not increased in SOD1 Mt MNs. DNA double strand breaks (DSBs) detected with anti-γH2AX and anti-53BP1 were indistinguishable in unstressed SOD Wt versus Mt (**A** row 1–2, **B**) or neither after induction of DSBs through 1 h treatment with 2 µM etoposide (**A** row 3–4, **C**) nor after 24 h-recovery after DSB induction with 5 µM etoposide (**A** row 5–6, **C**) in MAP2—positive human MNs. Similar results were observed using FACS analysis of γH2AX in MNs without stress and after DSB induction by X-ray irradiation with 2 Gy (**D**,**E**). Cell survival (**F**) was not affected by DNA damage induction with 2 µM etoposide in neither SOD1 Wt nor SOD1 Mt. SOD1 WT in image galleries (**A**): shown are representative examples of Ctrl1 (Table 1). SOD1 Mt in image galleries (**A**): shown are representative examples of D90A (Table 1). SOD WT pooled: data from Ctrl1 and Ctrl2 (**B**,**C**,**F**) or Ctrl2, Ctrl3 and Ctrl4 (**D**) were pooled (Table 1), SOD Mt pooled: data from R115G, A4V and D90A (**B**–**D**,**F**) (Table 1). Three (**B**,**C**,**F**) and four (**D**) independent biological replicates for each cell line and each condition. N.s.: no significant change in unpaired two-tailed Student’s *t*-test. Scale bar = 20 µm, in inlet = 5 µm.

**Figure 4 cells-11-01246-f004:**
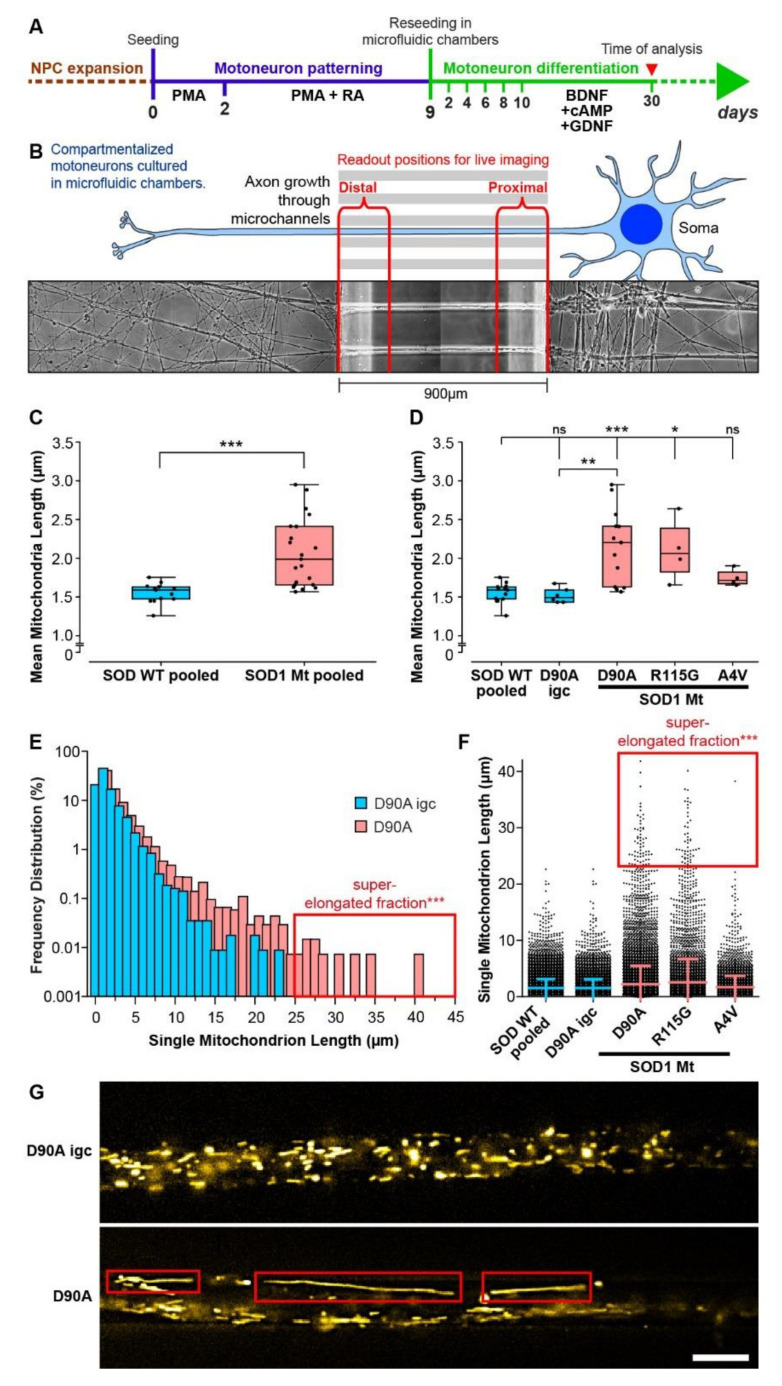
Increased elongation of mitochondria in mutant SOD1 spinal MNs. (**A**) Scheme of the differentiation pipeline to obtain iPSC-derived spinal motoneurons from patients, starting from cryopreservable neuronal precursor cells (NPCs). PMA: phorbol-12-myristate-13-acetate, RA: retinoic acid, BDNF: brain-derived neurotrophic factor, cAMP: cyclic adenosine monophosphate, GDNF: glia cell line-derived neurotrophic factor. (**B**) Cartoon of the live setup of MNs in microfluidic chambers (MFCs). The central microgroove of channels formed a physical barrier between the distal (left) and proximal (right) site where the somata were seeded. Only axons, not dendrites, could penetrate the microchannels. (**C**,**D**) Box plots of mean mitochondrial length values per experiment (black dots) for each respective line with 25–75% interquartile range (box), median (horizontal line) and non-outlier range (1.5-fold interquartile range added above and below to box, whiskers). SOD Wt pooled (**C**,**D**,**F**): data from all SOD Wt lines including D90A igc (*n* = 6) were pooled (*n* = 13) as listed in Table 1. SOD Mt pooled (C): data from D90A (*n* = 13), R115G (*n* = 4), and A4V (*n* = 4) (Table 1) were pooled (*n* = 21). Asterisks: highly significant increase in pairwise comparison of control cells (in blue) with each mutant line (in red), respectively, unpaired two-tailed Student’s *t*-test (**C**) or one-way ANOVA with Bonferroni post-hoc test (**D**). * *p* < 0.05, ** *p* < 0.01, *** *p* < 0.001, ns not significant. (**C**) Mitochondrial length was increased in mutant lines (SOD1 Mt) compared to wild type control cells (Ctrl). (**D**) Post hoc comparisons showed significantly longer mitochondria in mutant D90A and R115G but not in A4V. Comparison of D90A versus its isogenic gene-corrected control (D90A igc,) proved that this effect was due to the SOD1 mutation. (**E**) Length distribution of single mitochondria displayed as bar histogram in D90A (in red) compared to D90A igc (in blue), bin width 2µm. Note, the super-elongated fraction in D90A beyond 25 µm (red box) that was absent in D90A igc. (**F**) Length distribution of single mitochondria displayed as scatter dot plot (i.e., each dot presents one mitochondrion) for individual lines as in (**D**). Note, the super-elongated fraction present in D90A, R115G, and, to a lesser extent, A4V beyond 25 µm (red box) that were absent in both control conditions (pooled Ctrl, D90 igc). Center lines: medians, error bars: standard deviations. (**G**) Corresponding raw images obtained with mitotracker (yellow) showing representative examples of super-elongated mitochondria in D90A (red boxes) that were absent in D90A igc. Scale bar = 10 µm.

**Figure 5 cells-11-01246-f005:**
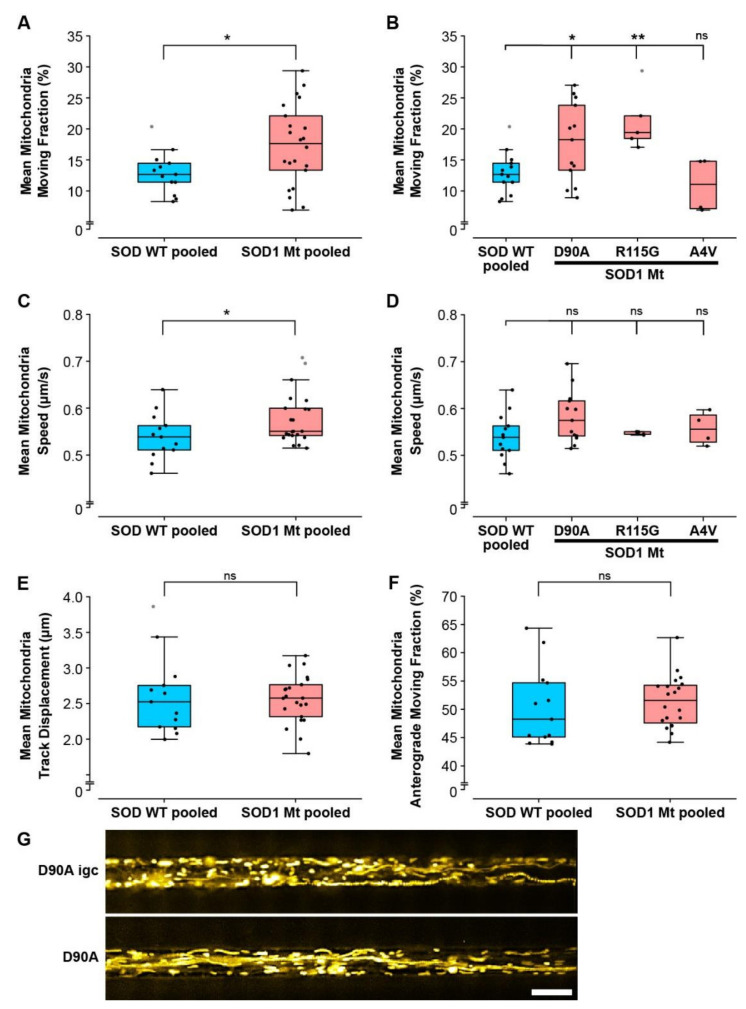
Altered motility of mitochondria in mutant SOD1 spinal MNs. (**A**–**F**) Box plots of various tracking parameters deduced from moving mitochondria as mean values per experiment (black dots) for each respective line with 25–75% interquartile range (box), median (horizontal line), non-outlier range (1.5-fold interquartile range added above and below to box, whiskers) and outliers (grey dots outside whiskers). SOD Wt pooled: data from all SOD Wt lines including D90A igc (*n* = 6) were pooled (*n* = 13) as listed in Table 1. SOD Mt pooled: data from D90A (*n* = 13), R115G (*n* = 4), and A4V (*n* = 4) (Table 1) were pooled (*n* = 21). Asterisks: highly significant increase in pairwise comparison of control cells (in blue) with mutant lines (in red), respectively, unpaired two-tailed Student’s *t*-test (**A**,**C**,**E**,**F**) or one-way ANOVA with Bonferroni post-hoc test (**B**,**D**). * *p* < 0.05, ** *p* < 0.01, n.s. not significant. (**A**) The percentage of moving mitochondria was increased in SOD1 Mt compared to SOD Wt pooled lines. (**B**) Comparison of single lines showed significantly higher percentages of moving mitochondria in D90A and R115G, but not in A4V. (**C**) Mean speed was significantly increased in pooled SOD1 Mt over Wt cells but single line comparisons (**D**) revealed no significant difference. (**E**) Track displacement of mitochondrial movement was not altered. (**F**) Percentage within the moving fraction as shown in (**A**) moving in anterograde direction towards distal axonal ends. The opposite retrograde direction towards somata results from the difference to 100%—anterograde fraction. (**G**) Maximum intensity projections of movie frame stacks obtained with mitotracker to illustrate mitochondria motility (yellow) in D90A igc versus D90A. Straight, processive movements are marked by long trajectories. Scale bar = 10 µm.

**Figure 6 cells-11-01246-f006:**
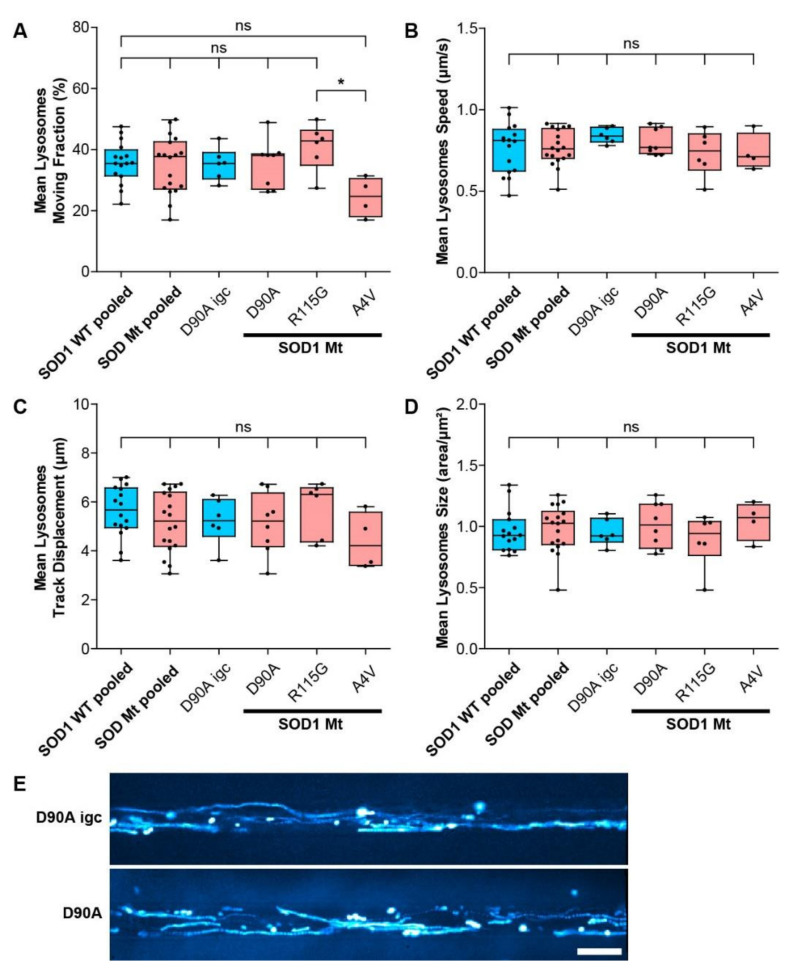
Lysosomal size and motility was not altered in mutant SOD1 spinal MNs. (**A**–**D**) Box plots of various tracking and morphological parameters deduced from moving lysosomes as mean values per experiment (black dots) for each respective line with 25–75% interquartile range (box), median (horizontal line) and non-outlier range (1.5-fold interquartile range added above and below to box, whiskers). SOD wildtype pooled: data from all SOD wildtypes lines including D90A igc (*n* = 6) were pooled (*n* = 13) as listed in Table 1. SOD mutants pooled: data from D90A (*n* = 13), R115G (*n* = 4), and A4V (*n* = 4) (Table 1) were pooled (*n* = 21). Ns: no significant change in any pairwise comparison of control cells (in blue) with mutant lines (in red), one-way ANOVA with Bonferroni post-hoc test. * *p* < 0.05, n.s. not significant. (**A**) Percentage of moving lysosomes, (**B**) lysosome mean speed, (**C**) track displacement, and (**D**) size in any SOD1 Mt lines was not altered compared to SOD1 WT cells including D90A igc. (**E**) Maximum intensity projections of movie frame stacks obtained with lysotracker to illustrate lysosome motility (cyan) in D90A igc versus D90A. Straight, processive movements are marked by long trajectories. Scale bar = 10 µm.

**Figure 7 cells-11-01246-f007:**
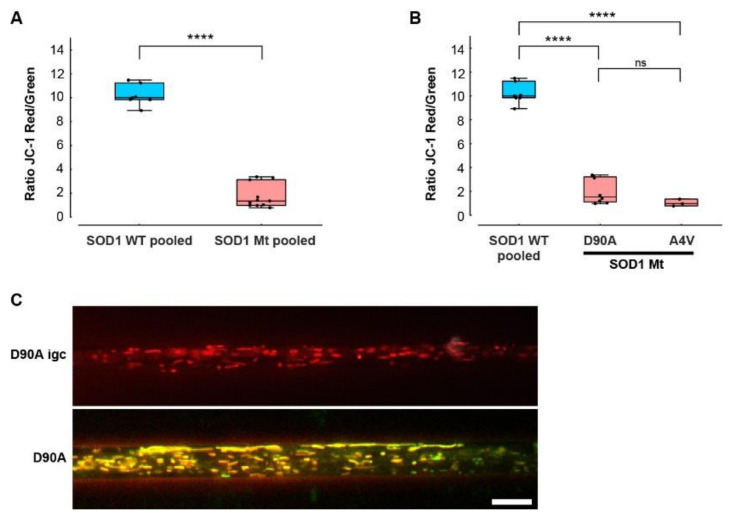
Reduced mitochondrial inner membrane potential in mutant SOD1 spinal MNs. (**A**,**B**) Box plots of membrane potential (fluorescence intensity ratio of JC1 red/green channel), mean values per experiment (black dots) for each respective line with 25–75% interquartile range (box), median (horizontal line), and non-outlier range (1.5-fold interquartile range added above and below to box, whiskers). SOD WT pooled: data from all SOD WT lines including D90A igc were pooled (*n* = 7) as listed in Table 1. SOD Mt pooled: data from D90A (*n* = 8) and A4V (*n* = 3) (Table 1) were pooled (*n* = 11). Asterisks: highly significant increase in pairwise comparison of control cells (in blue) with each mutant line (in red), respectively, unpaired two-tailed Student’s *t*-test (**A**) or one-way ANOVA with Bonferroni post test (**B**). **** *p* < 0.0001, n.s. not significant. (**A**) Membrane potential of mitochondria was strongly decreased in SOD1 Mt compared to SOD WT cells. (**B**) SOD1 A4V and D90A had both profoundly decreased mitochondrial membrane potentials. (**C**) Representative images of D90A igc versus D90A. Scale bar = 10 µm.

**Figure 8 cells-11-01246-f008:**
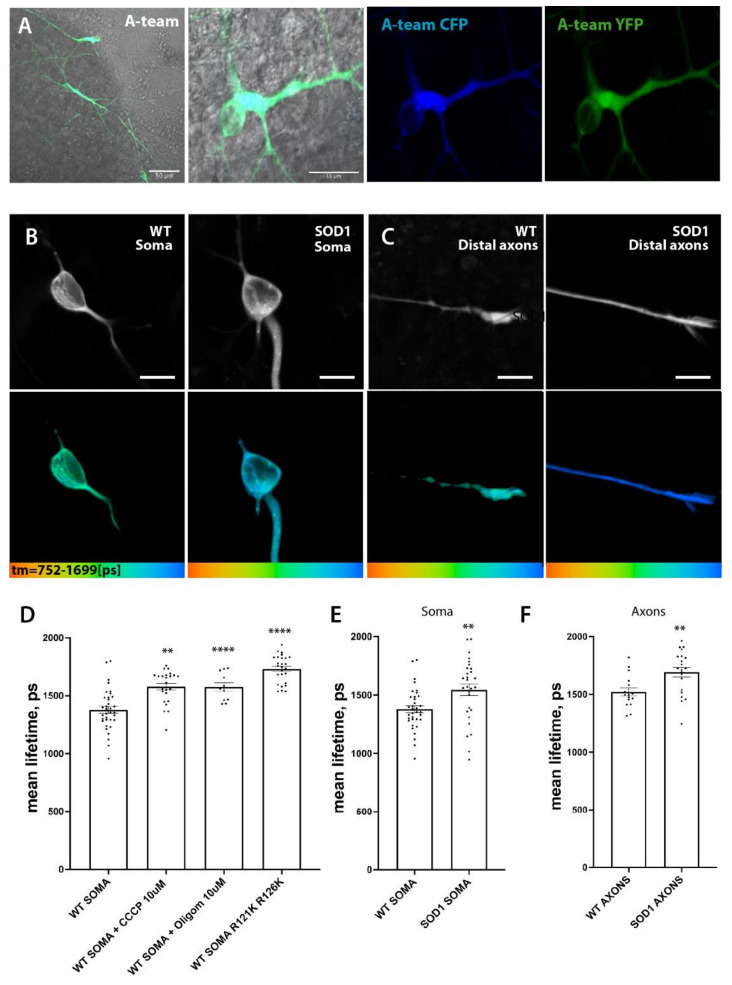
A-team FLIM reveals lower levels of ATP in SOD1 mutant soma and distant axons. (**A**) Low and higher magnification of the microfluidic chamber illustrating motoneurons lentivirally transfected with A-team FRET sensor. Individual subunits of the sensor could be visualized via their fluorescence in CFP and YFP channels. Low to moderate transfection efficiency allowed identifying single cells and thereby measurements of individual soma, proximal and also distal axonal (i.e., growth cones) compartments within each chamber. (**B**,**C**) Representative single images showing donor fluorescent intensity of ATP FRET sensor A-team and its corresponding color-coded range of mean-lifetime FLIM measurements within each cell. Images show different cell compartments, somata (**B**) and axons (**C**) in WT and SOD1 mutant MN cells. Color-coded scales of mean-lifetime values are shown at the bottom of the gallery. Scale bar is 10 µm. (**D**). Bar graphs comparing average mean lifetimes in untreated WT cells, WT cells treated with mitochondrial inhibitors CCCP (10 µM) and Oligomycin A (10 µM), as well as cells transfected with mutant non-sensing A-team (R121K R126K). Oxphos inhibition led to a significant increase in lifetime values (reduction in relative ATP concentration) in MNs. (**E**,**F**) Comparison of A-team mean lifetime values between WT and SOD1 showed a significant reduction of relative ATP concentration in both somata (**E**) and axons (**F**) of SOD1 mutant MN cells. Statistics unpaired two-tailed Student *t*-test, ** *p* < 0.01, **** *p* < 0.0001.

**Table 1 cells-11-01246-t001:** Patient/proband characteristics.

	SOD1 Line	Sex	Age at Biopsy	Mutation	Primarily Characterized in
Wt	Ctrl1	Female	48	-	[39]
Wt	Ctrl2	Female	43	-	[40]
Wt	Ctrl3	Female	49	-	[40]
Wt	Ctrl4	Male	34	-	[41]
IGC	SOD1 D90A igc	Female	46	-	[37]
Mt	SOD1 D90A	Female	46	D90A	[35]
Mt	SOD1 R115G	Male	59	R115G	[35]
Mt	SOD1 A4V	Female	73	A4V	

**Table 2 cells-11-01246-t002:** Summary of common phenotypes. (*): Revealed by GFP-A4V-SOD1 expression, not by immunostaining. (**): No Mock control with DMSO was required here because water was serving as stock solvent for arsenite, Ver und Pur. (↑): increase, (↓): decrease.

	Human Spinal Motor Neurons, DIV 30 (Endogenous SOD1 Expression Level)		
	R115G	D90A	4AV
**Mitochondrial elongation (Figure 4D–F)**	↑	↑	-
**Mitochondrial moving fraction (Figure 5B)**	↑	↑	-
**Mitochondrial inner membrane potential (Figure 7B)**	n.d.	↓	↓
**Lysosomal size (Figure 6D)**	-	-	-
**Lysosomal moving fraction (Figure 6A)**	-	-	↓
**DNA damage (Figure 3)**			
Mock (DMSO)	-	-	-
Etoposide	-	-	-
X-ray irradiation	-	-	-
**Total SOD1 in SG (Figure 1A,D *)**			
Untreated **	-	-	-
Arsenite	-	-	-
Heat stress	-	-	-
Ver + Puro			
**Misfolded SOD1 in SG (Figure 1A,C *)**			
Untreated **	-	-	-
Arsenite	-	-	-
Heat stress	-	-	-
Ver & Pur	-	-	-
**Total SOD1 in aggresome (Figure 1B)**			
Mock (DMSO)	-	-	-
0	-	-	-
**Misfolded SOD1 in aggresome (Figure 1B)**			
Mock (DMSO)	-	-	-
0	-	-	-
**Soluble total SOD1 (ELISA, Figure 2A)**			
Untreated	↓	↓	↓
Additional Ver & Pur	-	-	-
Additional MG132	-	-	-
**Soluble misfolded SOD1 (ELISA, Figure 2B)**			
Untreated	↑	-	↑
Additional Ver & Pur	-	-	-
Additional MG132	↑	-	↑
**Detergent-resistant SOD1 (WB, Figure 2C,D)**			
Untreated	-	-	-
Additional Ver & Pur	-	-	-
Additional MG132	↑	-	↑

## Data Availability

All data is presented in the manuscript.

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
