# Peer review of "Alteration of Mitochondrial Integrity as Upstream Event in the Pathophysiology of SOD1-ALS"

_cells, 2022, doi:10.3390/cells11071246_

Round 1

Reviewer 1 Report

Report for the authors

The manuscript entitled “Alteration of mitochondrial integrity as upstream event in the pathophysiology of SOD1-ALS”, submitted by Gunther R., Pal A. et al, describes an interesting study in which authors propose the hypothesis that mitochondria impairment may be a causative event of ALS disease.

Authors compare three mutant SOD1 models with wild type MNs to analyze different key steps in ALS pathology, such as DNA damage, stress granule induction and protein aggregation. In addition, they
investigate functions and morphology of mitochondria and lysosomes, obtaining that SOD1-mutants show altered mitochondrial dynamics and integrity. So, authors suggest that mitochondria may play a crucial role in mechanisms triggering SOD1-ALS phenotype. This result seems to be very promising in ALS research. However, I recommend major revision of this paper to proceed for publication.

First of all, the manuscript contains a lot of content and result to be rich in repetitions. The text is very difficult to understand. The introduction must be drastically shortened and you should follow the same scheme for M&M and Results.
Then, an extensive English revision and text editing is required.
Serious flaws need to be corrected before proceed to review the experimental study of the paper.

In addition, some of the gaps I found are reported below.
Table 1 contains 3 mutant sample, 4 wild type sample and 1 control line for D90A; however, in line 180, you report “three cell lines from healthy volunteers…”.
Line 182: last reference cited correspond to number 38. Later, in line, 211, you skipped to 42. References from 39 to 41 are completely missing.
Line 359-360: “Disordered SOD1 was quantified in freshly prepared extracts by misELISA (see below).”
Line 363-364: “Disordered SOD1 was quantified in cell extracts using a sandwich ELISA (misELISA), described previously”.
Final result: You didn’t describe the method.

Author Response

  1. The manuscript entitled “Alteration of mitochondrial integrity as upstream event in the pathophysiology of SOD1-ALS”, submitted by Gunther R., Pal A. et al, describes an interesting study in which authors propose the hypothesis that mitochondria impairment may be a causative event of ALS disease.

Authors compare three mutant SOD1 models with wild type MNs to analyze different key steps in ALS pathology, such as DNA damage, stress granule induction and protein aggregation. In addition, they
investigate functions and morphology of mitochondria and lysosomes, obtaining that SOD1-mutants show altered mitochondrial dynamics and integrity. So, authors suggest that mitochondria may play a crucial role in mechanisms triggering SOD1-ALS phenotype. This result seems to be very promising in ALS research. However, I recommend major revision of this paper to proceed for publication.

Response: We appreciate the efforts of the reviewer and the overall positive evaluation of our work. We have adressed all points raised by this reviewer in the revised manuscript, which we feel have contributed to a significant improvement of the text.

  1. First of all, the manuscript contains a lot of content and result to be rich in repetitions. The text is very difficult to understand. The introduction must be drastically shortened and you should follow the same scheme for M&M and Results.

Response: We appreciate that the text was overly long and repetitive, making it difficult to read in parts. The opinions of reviewer #1 and  #2 were quite disparate but in the revised manuscript we have strived to shorten and simplify the text whilst trying to maintain an appropriate level of detail and scientific rigour which should be both necessary and understandable for the interested reader. Since the introduction and discussion were already quite short, we feel that dramatically shortening these would have negated a proper introduction and discussion of the topic.However, we have strived to reduce these where possible. We have also simplified and streamlined the abstract..

We have also reduced the length oft he M&M – particualrly with reference to the SOD1 quantification methods, which are already published and described in detail. For other areas of the M&M, we believe that the interested reader will be eager to have an accurate depiction of the experimental setup as the combination of methods we have used here are rather unique. This should serve as a suitable resource for future experimentation in this field. In addition, reviewer #2 judged the research design as appropriate and the methods as adequately described. However, if the Editor thinks that is is appropriate, we are happy to move large portions of the M&M into the supplement.

  1. Then, an extensive English revision and text editing is required.

Response: We did another round of English language editing even thoug reviewer#2 already approved the English language.

  1. Serious flaws need to be corrected before proceed to review the experimental study of the paper.

Response: We have address all specific comments from the reviewer. However, we do not agree that our study shows serious methodological flaws. We have been extremely rigarous in our experimental design and out attempts to quantify changes in mitochondria in relation to previously described pathological changes on ALS MNs. In this respect we believe out study is somewhat unique in its openess and integrity.

  1. In addition, some of the gaps I found are reported below.
    Table 1 contains 3 mutant sample, 4 wild type sample and 1 control line for D90A; however, in line 180, you report “three cell lines from healthy volunteers…”.

Response. We thank the reviewer for pointing out this omission and corrected this in the revised manuscript..

  1. Line 182: last reference cited correspond to number 38. Later, in line, 211, you skipped to 42. References from 39 to 41 are completely missing.

Response. We would like to draw the reviewers attention to the table 1, in which the „lacking“ references are included. Thus the sequence of citations is in line with the journals requirements.

  1. Line 359-360: “Disordered SOD1 was quantified in freshly prepared extracts by misELISA (see below).”
    Line 363-364: “Disordered SOD1 was quantified in cell extracts using a sandwich ELISA (misELISA), described previously”.

Response. This part was removed during the shortening.

  1. Final result: You didn’t describe the method.

Response: SInce the SOD1 quantification methods have been extensicely described in published papers, we have shortened this part of the methods in the revised version and referred to these papers in the text.

Reviewer 2 Report

Gunther et al used a human iPSC model of spinal MNs and three different endogenous ALS-associated SOD1 mutations (D90Ahom, R115Ghet or A4Vhet) to investigate early cellular changes. They identified loss of mitochondrial, but not lysosomal, integrity as the earliest common pathological phenotype among different SOD1 mutations. Altered morphology with super-elongated mitochondria and impaired inner mitochondrial membrane potential was a common feature in mutant SOD1 iPSC-derived MNs. Misfolding and aggregation of SOD1 could be induced by inhibiting the proteasome but not via the stress granule pathway. Mitochondrial homeostasis is necessary for the vitality and viability of neuronal cells.

These data suggest that mitochondrial dysfunction is one of the first crucial steps in the pathogenic cascade that leads to SOD1-ALS.

The authors showed that impaired mitochondrial integrity was most prominent in MNs D90Ahom mutation, whereas both soluble disordered and detergent-resistant misfolded SOD1 was more prominent in R115G and A4V mutant lines. These results highlight the need for individualized medical approaches for SOD1-ALS. Based on the data, the authors should address the homozygous recessive mutation of D90A versus heterozygous dominant R115G and A4V in a more cohesive manner.

Lines 55-56: The sentence should be corrected:  “…the clinical relevance of SOD1 overexpression models is questionable.”

Author Response

Gunther et al used a human iPSC model of spinal MNs and three different endogenous ALS-associated SOD1 mutations (D90Ahom, R115Ghet or A4Vhet) to investigate early cellular changes. They identified loss of mitochondrial, but not lysosomal, integrity as the earliest common pathological phenotype among different SOD1 mutations. Altered morphology with super-elongated mitochondria and impaired inner mitochondrial membrane potential was a common feature in mutant SOD1 iPSC-derived MNs. Misfolding and aggregation of SOD1 could be induced by inhibiting the proteasome but not via the stress granule pathway. Mitochondrial homeostasis is necessary for the vitality and viability of neuronal cells.

These data suggest that mitochondrial dysfunction is one of the first crucial steps in the pathogenic cascade that leads to SOD1-ALS.

The authors showed that impaired mitochondrial integrity was most prominent in MNs D90Ahom mutation, whereas both soluble disordered and detergent-resistant misfolded SOD1 was more prominent in R115G and A4V mutant lines. These results highlight the need for individualized medical approaches for SOD1-ALS. Based on the data, the authors should address the homozygous recessive mutation of D90A versus heterozygous dominant R115G and A4V in a more cohesive manner.

Response: We would like to thank the reviewer for the very positive overall review. We have presented the data of the homozygous vs. heterozygous mutants more cohesively.

Lines 55-56: The sentence should be corrected:  “…the clinical relevance of SOD1 overexpression models is questionable.”

Response. We rewrote this sentence.

Round 2

Reviewer 1 Report

Thanks for the deep revision of the manuscript. In my opinion, the work has been greatly improved.

Author Response

We deeply thank the reviewer for accepting the manuscript!

Reviewer 2 Report

The suggestions have been complied.

Author Response

(The authors gave the same response as above.)
